# MULTI-PRECISION POLICY ENFORCED TRAINING (MUPPET) : A PRECISION-SWITCHING STRATEGY FOR QUANTISED FIXED-POINT TRAINING OF CNNS

## ABSTRACT

Large-scale convolutional neural networks (CNNs) suffer from very long training times, spanning from hours to weeks, limiting the productivity and experimentation of deep learning practitioners. As networks grow in size and complexity, training time is reduced through low-precision data representations and computations. However, in doing so the final accuracy suffers due to the problem of vanishing gradients. Existing state-of-the-art methods combat this issue by means of a mixed-precision approach utilising two different precision levels, FP32 (32-bit floating-point precision) and FP16/FP8 (16-/8-bit floating-point precision), leveraging the hardware support of recent GPU architectures for FP16 operations to obtain performance gains.This work pushes the boundary of quantised training by employing a multilevel optimisation approach that utilises multiple precisions including low-precision fixed-point representations. The novel training strategy, MuPPET, combines the use of multiple number representation regimes together with a precision-switching mechanism that decides at run time the transition point between precision regimes. Overall, the proposed strategy tailors the training process to the hardware-level capabilities of the utilised hardware architecture and yields improvements in training time and energy efficiency compared to state-of-the-art approaches. Applying MuPPET on the training of AlexNet, ResNet18 and GoogLeNet on ImageNet (ILSVRC12) and targeting an NVIDIA Turing GPU, MuPPET achieves the same accuracy as standard full-precision training with an average training-time speedup of $1.28\times$ across the networks.

## 1 INTRODUCTION

Convolutional neural networks (CNNs) have demonstrated unprecedented accuracy in various machine learning tasks, from video understanding (Gan et al., 2015; He et al., 2018) to drone navigation (Loquercio et al., 2018; Kouris & Bouganis, 2018). To achieve such high levels of accuracy in inherently complex applications, current methodologies employ the design of large and complex CNN models (Szegedy et al., 2017; Huang et al., 2017) trained over large datasets (Deng et al., 2009; Lin et al., 2014). Nevertheless, the combination of large models and massive datasets results in long training times. This in turn leads to long turn-around times which limits the productivity of deep learning practitioners and prohibits wider experimentation. For instance, automatic tuning and search of neural architectures (Cai et al., 2018; Zhong et al., 2018) is a rapidly advancing area where accelerated training enables improving the produced networks.

To counteract these long turn-around times, substantial research effort has been invested in hyperparameter tuning for the acceleration of training, with a particular focus on batch size and content. One line of work maximises memory and hardware utilisation by changing the batch size, in order to perform CNN training specific prefetching, scheduling and dependency improvement (Chen et al., 2019; Rhu et al., 2016). Other works focus on altering batch size or reconstructing minibatches to improve the convergence rate while sustaining high hardware utilisation (Devarakonda et al., 2017; Johnson & Guestrin, 2018; Peng et al., 2019), demonstrating up to $6.25\times$ training speedup.

A number of studies have focused on the use of reduced-precision training schemes. Reduced-precision arithmetic involves the utilisation of data formats that have smaller wordlengths than the conventional 32-bit floating-point (FP32) representation and is an approach for co-optimising pro-

cessing speed, memory footprint and communication overhead. Existing literature can be categorised into works that use reduced precision to accelerate only the training stage while targeting an FP32 model, and those that produce networks with quantised weights. Regarding the former, Courbariaux et al. (2015) and Gupta et al. (2015) utilise dynamic quantisation and stochastic rounding respectively as a means to combat the accuracy loss due to quantisation.

Nevertheless, the effectiveness of the proposed schemes have only been demonstrated on small-scale datasets such as CIFAR-10 and MNIST, and on a limited set of networks. Furthermore, the range of quantisation levels that has been explored varies greatly, with a number of works attacking the problem by focusing on mild quantisation levels such as half-precision floating-point (FP16) (Micikevicius et al., 2018), while others focus on lower precisions such as 8-bit floating-point (FP8) (Wang et al., 2018). Finally, quantisation has also been used as a means of reducing the memory and communication overhead in distributed training (De Sa et al., 2015; 2017; Alistarh et al., 2017).

At the same time, the characteristics of modern CNN workloads and the trend towards quantised models have led to an emergence of specialised hardware processors, with support for low-precision arithmetic at the hardware level. From custom designs such as Google's TPUs (Jouppi et al., 2017) and Microsoft's FPGA-based Brainwave system (Fowers et al., 2018) to commodity devices such as NVIDIA's Turing GPUs, existing platforms offer native support for reduced-precision data types including 16-bit floating-point (FP16), and 8- (INT8) and 4-bit (INT4) fixed-point, providing increased parallelism for lower bitwidths. Although these platforms have been mainly designed for the inference stage, the low-precision hardware offers significant opportunities for accelerating the time-consuming training stage. In this respect, there is an emerging need to provide training algorithms that can leverage these existing hardware optimisations and provide higher training speed.

This work tackles the field of reduced-precision training at an algorithmic level. Independently of the number of quantisation levels chosen, or how extreme the quantisation is, this work proposes a metric that estimates the amount of information each new training step obtains for a given quantisation level, by capturing the diversity of the computed gradients *across epochs*. This enables the design of a policy that, given a set of quantisation levels, decides *at run time* appropriate points to increase the precision of the training process at that current instant without impacting the achieved test accuracy compared to training in FP32. Due to its agnostic nature, it remains orthogonal and complementary to existing low-precision training schemes. Furthermore, by pushing the precision below the 16-bit bitwidth of existing state-of-the-art techniques, the proposed method is able to leverage the low-precision capabilities of modern processing systems to yield training speedups without penalising the resulting accuracy, significantly improving the time-to-accuracy trade-off.

## 2 BACKGROUND AND RELATED WORK

The state-of-the-art method in training in reduced precision is mixed-precision training (Micikevicius et al., 2018). The authors propose to employ low-precision FP16 computations in the training stage of high-precision CNNs that perform inference in FP32. Along the training phase, the algorithm maintains a high-precision FP32 copy of the weights of the network, known as a *master copy*. At each minibatch, the inputs and weights are quantised to FP16 with all computations of the forward and backward pass performed in FP16, yielding memory footprint and runtime savings. Under this scheme, each stochastic gradient descent (SGD) update step entails accumulating FP16 gradients into the FP32 master copy of the weights, with this process performed iteratively throughout the training of the network. Micikevicius et al. (2018) evaluate their scheme over a set of state-of-the-art models on ImageNet, and show that mixed-precision training with FP16 computations achieves comparable accuracy to standard FP32 training.

Wang et al. (2018) also presented a method to train an FP32 model using 8-bit floating-point (FP8). The authors propose a hand-crafted FP8 data type, together with a chunk-based computation technique, and employ strategies such as stochastic rounding to alleviate the accuracy loss due to training at reduced precision. For AlexNet, ResNet18 and ResNet50 on ImageNet, Wang et al. (2018) demonstrates comparable accuracy to FP32 training while performing computations in FP8.

Additionally the works presented in (Zhou et al., 2016; Chen et al., 2017) approach the problem of reduced-precision training employing fixed-point computations. FxpNet (Chen et al., 2017) was only evaluated on CIFAR-10, failing to demonstrate performance on more complex datasets such as ImageNet. DoReFa-net (Zhou et al., 2016) was tested on ImageNet but only ran on AlexNet missing out on state-of-the-art networks such as GoogLeNet and ResNet.

All related works focus on accelerating the training of an FP32 model through reduced-precision computations. At the hardware level, 8-bit fixed-point multiplication uses $18.5\times$ less energy and $27.5\times$ less area with up to $4\times$ lower multiplication times than FP32 (Sze et al., 2017). Consequently, this work attempts to push the boundaries of reduced-precision training by moving to reduced-precision fixed-point computations while updating an FP32 model.

Preliminary tests (Sec. 4.4 for details) demonstrated that training solely in 8-bit fixed-point results in a significant degradation of validation accuracy compared to full FP32 training. This work aims to counteract this degradation by progressively increasing the precision of computations throughout training in an online manner determined by the proposed metric inspired by gradient diversity (Yin et al., 2018). Additionally by operating in an online fashion, MuPPET tailors the training process to best suit the particular network-dataset pair at each stage of the training process.

Gradient diversity was introduced by Yin et al. (2018) as a metric of measuring the dissimilarity between sets of gradients that correspond to different minibatches. The gradient diversity of a set of gradients is defined as

$$\Delta_{\mathcal{S}}(\mathbf{w}) = \frac{\sum_{i=1}^n ||\nabla f_i(\mathbf{w})||_2^2}{||\sum_{i=1}^n \nabla f_i(\mathbf{w})||_2^2} = \frac{\sum_{i=1}^n ||\nabla f_i(\mathbf{w})||_2^2}{\sum_{i=1}^n ||\nabla f_i(\mathbf{w})||_2^2 + \sum_{i\neq j}\langle \nabla f_i(\mathbf{w}), \nabla f_j(\mathbf{w})\rangle} \quad (1)$$

where $\nabla f_i(\mathbf{w})$ represents the gradient of weights $\mathbf{w}$ for minibatch $i$.

The key point to note in Eq. (1) is that the denominator contains the inner product between two gradients from *different minibatches*. Thus, orthogonal gradients would result in high gradient diversity, while similar gradients would result in low gradient diversity. The proposed framework, MuPPET, enhances this concept by considering gradients between minibatches *across epochs* and proposes the developed metric as a proxy for the amount of new information gained in each training step. Section 3 further expands on how gradient diversity is incorporated into the MuPPET algorithm.

## 3 METHODOLOGY

### 3.1 MULTILEVEL OPTIMISATION FOR TRAINING CNNS

Conventionally, the training process of a CNN can be expressed as in Eq. (2). Given a CNN model $f$ parameterised by a set of weights $\mathbf{w} \in \mathbb{R}^D$, where $D$ is the number of weights of $f$, training involves a search for weight values that minimise the task-specific empirical loss, $Loss$, on the target dataset. Typically, a fixed arithmetic precision is employed across the training algorithm with FP32 currently being the *de facto* representation used by the deep learning community.

$$\min_{\mathbf{w}^{(\text{FP32})} \in \mathbb{R}^D} Loss(f(\mathbf{w}^{(\text{FP32})})) \quad (2)$$

The proposed method follows a different approach by introducing a multilevel optimisation scheme (Migdalas et al., 2013) that leverages the performance gains of reduced-precision arithmetic. The single optimisation problem of Eq. (2) is transformed into a series of optimisation problems with each one employing different precision for computations, but maintaining weights storage at FP32 precision. Under this scheme, an $N$-level formulation comprises $N$ sequential optimisation problems to be solved, with each level corresponding to a "finer" model.

Overall, this formulation adds a hierarchical structure to the training stage, with an increasing arithmetic precision across the hierarchy of optimisation problems. Starting from the $N$-th problem, the inputs, weights, and activations of the CNN model $f$ are quantised with precision $q^N$, which is the lowest precision in the system and represents the coarsest version of the model. Each of the $N$ levels progressively employs higher precision until the first level is reached, which corresponds to the original problem of Eq. (2). Formally, at the i-th level, the optimisation problem is formulated as

$$\min_{\mathbf{w}^{(q^i)} \in \mathcal{V}} Loss(f(\mathbf{w}^{(q^i)})) \quad \text{s.t. } \mathcal{V} = \left\{ \mathbf{w}^{(q^i)} \in [\text{LB}, \text{UB}]^D \right\} \quad (3)$$

where LB and UB are the lower and upper bound in the representational range of precision $q^i$. The target CNN model $f$ uses a set of weights quantised with precision $q^i$ and hence the solution of this optimisation problem can be interpreted as an approximation to the original problem of Eq. (2). To transition from one level to the next, the result of each level of optimisation is employed as a starting point for the next level, up to the final outermost optimisation that reduces to Eq. (2).

## 3.2 THE MUPPET ALGORITHM

Fig. 1 presents the process of train-
ing a CNN using the proposed al-
gorithm. All figures in this pa-
per are shown in the Appendix B
at a larger scale for enhanced read-
ability. Within each epoch, MuP-
PET performs mixed-precision train-
ing where the weights are stored in
an FP32 master copy and are quan-
tised to the desired fixed-point preci-
sion on-the-fly. At epoch $j$, the com-
putations for the forward and back-
ward passes ($F$ and $B$ blocks respec-
tively) are performed at the current
quantised precision ($q^j$) and the ac-

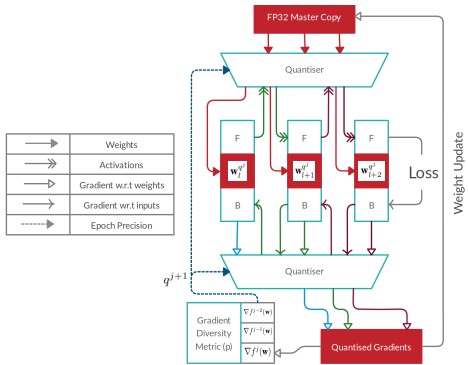

Figure 1: Precision-switching training
scheme of MuPPET.

tivations as well as the gradients obtained from each layer are quantised by the quantiser module
before being passed on to the next layer, or stored. After each minibatch, the full-precision mas-
ter copy of the weights is updated using a quantised gradient matrix. As discussed in Section 3.1,
the quantisation level is gradually increased over the period of the training. In MuPPET, switching
between these optimisation levels at the correct times is crucial in order not to compromise the fi-
nal validation accuracy. In this respect, MuPPET introduces a precision switching policy based on
an inter-epoch gradient diversity (Yin et al., 2018) metric that dictates when to switch to the next
precision. Details of the switching policy are presented in Section 3.3.

### 3.2.1 QUANTISATION STRATEGY

In order to implement quantised training, a quantisation strategy needs to be defined. The proposed
dynamic quantisation strategy utilises block floating-point arithmetic (also known as dynamic fixed-
point), where each fixed-point number is represented as a pair of an $\text{WL}^{\text{net}}$-bit signed integer $x$ and
a scale factor $s$, such that the value is represented as $x \times 2^{-s}$.

During the forward and backward passes of the training process, the weights and feature maps are
both quantised, and the multiplication operations are performed at the same low precision. The
quantisation method employs a stochastic rounding methodology (Gupta et al., 2015). The accu-
mulation stage of the matrix-multiply operation is accumulated into a 32-bit fixed-point value to
prevent overflow on the targeted networks.[1] The result of this matrix multiplication is subsequently
quantised to the target wordlength before being passed as input to the next layer. Following the
block floating-point scheme, quantisation is performed such that each weight and feature map ma-
trix in the network has a single scale factor shared by all values within the matrix. The quantisation
configuration for the i-th level of optimisation and the l-th layer, $q_l^i$, and the full set of configurations,
$q^i$, are given by left- and right-hand side of Eq. (4) respectively.

$$q_l^i = \left\langle \text{WL}^{\text{net}}, s_l^{\text{weights}}, s_l^{\text{act}} \right\rangle^i, \quad \forall l \in [1, |\mathcal{L}|] \quad \text{and} \quad q^i = \left\langle q_l^i \mid \forall l \in [1, |\mathcal{L}|] \right\rangle \tag{4}$$

where $|\mathcal{L}|$ is the number of layers of the target network, $\text{WL}^{\text{net}}$ is the fixed wordlength across the
network, $s_l^{\text{weights}}$ and $s_l^{\text{act}}$ are the scaling factors for the weights and activations respectively, of the l-
th layer for the i-th level of optimisation. As a result, for $N$ levels, there are $N$ distinct quantisation
schemes; $N - 1$ of these schemes are with varying fixed-point precisions, and the finest level of
quantisation, $q^1$, is single-precision floating-point (FP32). The scaling factor for a matrix $\mathbf{X}$ is first
calculated as shown in Eq. (5) and individual elements are quantised as in Eq. (6).

$$s^{\{\text{weights, act}\}} = \left\lfloor \log_2 \left( \min \left( \frac{\text{UB} + 0.5}{\mathbf{X}_{\max}^{\{\text{weights, act}\}}}, \frac{\text{LB} - 0.5}{\mathbf{X}_{\min}^{\{\text{weights, act}\}}} \right) \right) \right\rfloor \tag{5}$$

$$x_{\text{quant}}^{\{\text{weights, act}\}} = \left\lfloor x^{\{\text{weights, act}\}} \cdot 2^{s^{\{\text{weights, act}\}}} + \text{Unif}\left(-0.5, 0.5\right) \right\rceil \tag{6}$$

---

[1]The accumulator wordlength is large enough to accommodate the current CNN models, without overflow.

where $\mathbf{X}^{\{\text{weights, act}\}}_{\{\text{max, min}\}}$ is either the maximum or minimum value in the weights or feature maps matrix of the current layer, LB and UB are the lower and upper bound of the current wordlength $\text{WL}^{\text{net}}$, and Unif(a,b) represents sampling from the uniform distribution in the range [a,b]. Eq. (5) adds $0.5$ and $-0.5$ to UB and LB respectively to ensure maximum utilisation of $\text{WL}^{\text{net}}$.

### 3.2.2 INFORMATION TRANSFER BETWEEN LEVELS

Employing multilevel training for CNNs requires an appropriate mechanism for transferring information between levels. To achieve this, the proposed optimiser maintains a master copy of the weights in full precision (FP32) throughout the optimisation levels. Similar to mixed-precision training (Micikevicius et al., 2018), at each level the SGD update step is performed by accumulating a fixed-point gradient value into the FP32 master copy of the weights. Starting from the coarsest quantisation level $i = N$, to transfer the solution from level $i$ to level $i - 1$, the master copy is quantised using the quantisation scheme $q^{i-1}$. With this approach, the weights are maintained in FP32 and are quantised on-the-fly during run time in order to be utilised in each training step.

### 3.3 PRECISION SWITCHING POLICY

The metric to decide when to switch between levels of quantisation is inspired by Yin et al. (2018) and based on the concept of gradient diversity (Eq. (1)). MuPPET computes $\Delta_{\mathcal{S}}(\mathbf{w})$ between gradients obtained across epochs as a proxy to measure the information that is obtained during the training process; the lower the diversity between the gradients, the less information this level of quantisation provides towards the training of the model. Therefore, the proposed method comprises a novel normalised inter-epoch version of the gradient diversity along with a run-time policy to determine the epochs to switch precision.

The following policy is employed to determine when a precision switch is to be performed. For a network with layers $\mathcal{L}$ and a quantisation scheme $q^i$ that was switched into at epoch $e$:

1. For each epoch $j$ and each layer $l \in \mathcal{L}$, the last minibatch's gradient, $\nabla f_l^j(\mathbf{w})$, is stored.

2. After $r$ (resolution) number of epochs, the inter-epoch gradient diversity at epoch $j$ is

$$\Delta_{\mathcal{S}}(\mathbf{w})^j = \frac{\sum_{\forall l \in \mathcal{L}} \frac{\sum_{k=j-r}^{j} ||\nabla f_l^k(\mathbf{w})||_2^2}{|| \sum_{k=j-r}^{j} \nabla f_l^k(\mathbf{w})||_2^2}}{|\mathcal{L}|} \tag{7}$$

3. At an epoch $j$, given a set of gradient diversities $\mathcal{S}(j) = \left\{ \Delta_{\mathcal{S}}(\mathbf{w})^i \quad \forall \ e \leq i < j \right\}$, the ratio $p = \frac{\max \mathcal{S}(j)}{\Delta_{\mathcal{S}}(\mathbf{w})^j}$ is calculated.

4. An empirically determined decaying threshold $T = \alpha + \beta e^{-\lambda j}$ (8) is placed on the ratio $p$.

5. If the $p$ violates $T$ more than $\gamma$ times, a precision switch is triggered and $\mathcal{S}(j) = \emptyset$.

As long as the gradients across epochs remain diverse, $\Delta_{\mathcal{S}}(\mathbf{w})^j$ (Eq.(7)) at the denominator of $p$ sustains a high value and the value of $p$ remains low. However, when the gradients across epochs become similar, $\Delta_{\mathcal{S}}(\mathbf{w})^j$ decreases and the value of $p$ becomes larger. *Generalisability across epochs* is obtained as $p$ accounts for the change in information relative to the maximum information available since the last precision change. Hence, the metric acknowledges the presence of temporal variations in information provided by the gradients. *Generalisability across networks* and *datasets* is maintained as $p$ measures a ratio. Consequently, the absolute values of gradients which could vary between networks and datasets, matter less. Overall, MuPPET employs the metric $p$ as a mechanism to trigger a precision switch whenever $p$ violates threshold $T$ more than $\gamma$ times.

The likelihood of observing $r$ gradients across $r$ epochs that have low gradient diversity, especially at early stages of training is low. The intuition applied here is that when this does happen at a given precision, it may be an indication that information is being lost due to quantisation and thus corresponds to a high $p$ value, which argues to move to a higher bitwidth.

### 3.3.1 HYPERPARAMETERS

The hyperparameters for the proposed MuPPET algorithm are the following: 1) values of $\alpha$, $\beta$, and $\lambda$ that define the decaying threshold from Eq. (8), 2) the number of threshold violations allowed

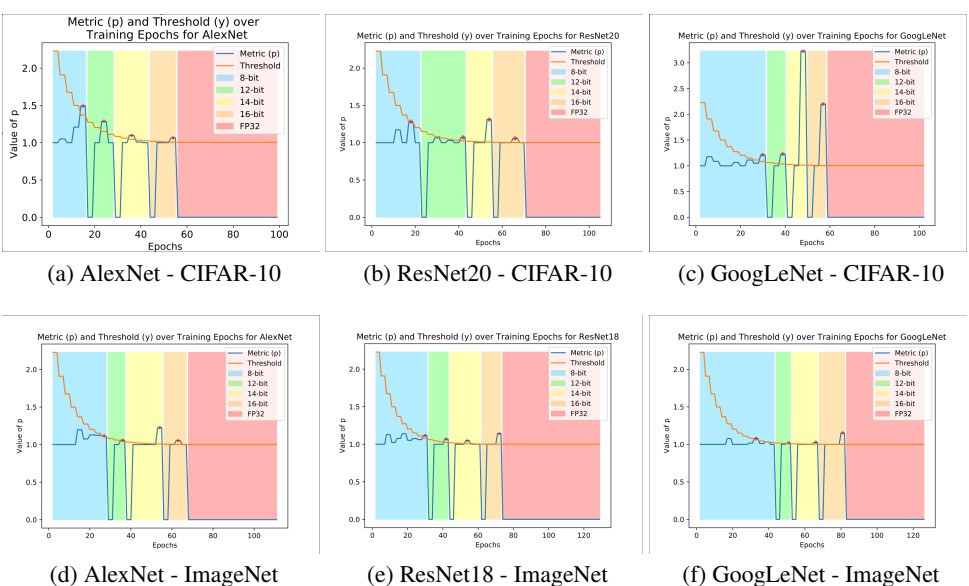

Figure 2: Demonstration of the generalisability of $p$ over networks, datasets and epochs.

before the precision change is triggered ($\gamma$), 3) the resolution $r$, 4) the set of precisions at which training is performed, and 5) the epochs at which the learning rate is changed. The values of $\alpha$, $\beta$, $\lambda$, $r$, and $\gamma$ were set at 1, 1.5, 0.1, 3, and 2 respectively after empirical cross-validation. These were tuned by running training on AlexNet and ResNet20 on the CIFAR-10 dataset. All MuPPET hyperparameters remain the same regardless of network or dataset. Regarding training hyperparameters, batch size was increased from 128 to 256 going from CIFAR-10 to ImageNet. All other training hyperparameters, including learning rate remained constant. Analysis of generalisability and the training hyperparameters used are presented in Section 4.1. The empirically-chosen quantised precisions at which training was performed were 8-, 12-, 14- and 16-bit fixed-point. Precisions below this did not result in any progress towards convergence for any network.

Overall, MuPPET introduces a policy that allows to decide at run time an appropriate point to switch between quantisation levels. After training at 16-bit fixed-point, the rest of the training is performed at FP32 until the desired validation accuracy is reached. Decaying the learning rate causes a finer exploration of the optimisation space as does increasing the quantisation level. Therefore, the learning rate was kept constant during quantised training and was decayed only after switching to FP32.

## 4 EVALUATION OF MUPPET

### 4.1 GENERALISABILITY

The MuPPET framework was evaluated on its applicability across epochs, networks and datasets. Fig. 2 shows the value of the metric $p$ over the epochs in blue, and the decaying threshold described in Eq. (8) in orange. The number of epochs for which training in each precision was performed is shown by the various overlay colours. The first violation is denoted by a red dot and the second violation is not seen as it occurs exactly at the point of switching. The graphs show that across various networks and datasets, the values of $p$ stay relatively similar, backing the choice of a universal decaying factor. Furthermore, empirical results for CIFAR-10 indicated that changing from one fixed-point precision to another too early in the training process had a negative impact on the final validation accuracy. Using a decaying threshold ensures that the value of $p$ needs to be much higher in the initial epochs to trigger a precision change due to the volatility of $p$ in early epochs of training.

### 4.2 PERFORMANCE EVALUATION

The accuracy results presented in this section utilised the proposed stochastic quantisation strategy. The methodology was developed using PyTorch. As the framework does not natively support low-precision implementations, all quantisation and computations corresponding to 8-, 12-, 14-, and 16-bit precisions were performed through emulation on floating-point hardware. All hyperparameters

not specified below were left as PyTorch defaults. For all networks, an SGD optimiser was used with batch sizes 128 on CIFAR-10 or 256 on ImageNet, momentum of 0.9 and weight decay of $1e^{-4}$.

As a baseline, an FP32 model with identical hyperparameters (except for batch size) was trained. The baseline FP32 training was performed by training for 150 epochs and reducing the learning rate by a factor of 10 at epochs 50 and 100. In order to achieve comparable final validation accuracy to the FP32 baseline, once MuPPET triggered a precision change out of 16-bit fixed-point, 45 training epochs at FP32 precision were performed. The learning rate was reduced by a factor of 10 every 15 FP32 training epochs. For AlexNet, ResNet18, ResNet20, and GoogLeNet, the initial learning rate was set to 0.01, 0.1, 0.1, and 0.001 respectively. The detailed breakdown of the ImageNet training runs with training and validation loss curves can be found in the Appendix A.

| | CIFAR-10 | | | ImageNet | | |
|---|---|---|---|---|---|---|
| | FP32 | MuPPET | Diff (pp) | FP32 | MuPPET | Diff (pp) |
| **AlexNet** | 75.45 | 74.49 | -0.96 | 56.21 | 55.33 | -0.88 |
| **ResNet** | 90.08 | 90.86 | 0.78 | 69.48 | 69.09 | -0.39 |
| **GoogLeNet** | 89.23 | 89.47 | 0.24 | 59.15 | 63.70 | 4.55 |

Table 1: Top-1 validation accuracy (%) on CIFAR-10 and ImageNet for FP32 baseline and MuPPET

Table 1 presents the achieved Top-1 validation accuracy of MuPPET and the FP32 baseline, together with the accuracy difference in percentage points (pp). As shown on the table, MuPPET is able to provide comparable Top-1 validation accuracy to standard FP32 training across both networks and datasets. Due to a sub-optimal training setup of GoogLeNet on ImageNet, the baseline and MuPPET training severely underperformed compared to the reported state-of-the-art works. Nevertheless, the results demonstrate the quality of training with MuPPET using identical hyperparameters. As a result, MuPPET's performance demonstrates the effectiveness of the precision switching strategy in achieving significant acceleration of training time (Section 4.3) at negligible cost in accuracy by running many epochs at lower precision, particularly on very large datasets.

## 4.3 WALL-CLOCK TIME IMPROVEMENTS

This section explores the gains in estimated wall-clock time of the current implementation of MuPPET (Current Impl.) with respect to baseline FP32 training, Mixed Precision by Micikevicius et al. (2018) and MuPPET's ideal implementation (Table 2). For all performance results, the target platform was an NVIDIA RTX 2080 Ti GPU. At the moment, deep learning frameworks, such as PyTorch, do not provide native support for reduced-precision hardware. Consequently, the wall-clock times in Table 2 were estimated using a performance model developed with NVIDIA's CUTLASS library (Kerr et al., 2018) for reduced-precision general matrix-multiplication (GEMM) employing the latest Turing architecture GPUs. The GEMMs that were accelerated were in the convolutional and fully-connected layers of each network. INT8 hardware was used to profile the 8-bit fixed-point computations, while FP16 hardware was used to profile 12-, 14-, and 16-bit fixed-point computations as well as Mixed Precision (Micikevicius et al., 2018) wall-clock time. CUTLASS (Kerr et al., 2018) natively implements bit-packing to capitalise on improved memory-bandwidth utilisation. The model for the current implementation is limited by the fact that frameworks force quantisation to happen to and from FP32. For the MuPPET (Ideal) scenario, the model assumes native hardware utilisation which would reduce the overhead by removing this restriction.

As shown on Table 2, MuPPET consistently achieves $1.25$-$1.32\times$ speedup over the FP32 baseline across the networks when targeting ImageNet on the given GPU. With respect to Mixed Precision, the proposed method outperforms it on AlexNet by $1.23\times$ and delivers comparable performance for ResNet18 and GoogLeNet. Currently, the absence of native quantisation support, and hence the necessity to emulate quantisation and the associated overheads, is the limiting factor for MuPPET to achieve higher processing speed. In this respect, MuPPET run on native hardware would yield $1.05\times$ and $1.48\times$ speedup for ResNet18 and GoogLeNet respectively compared to Mixed Precision. As a result, MuPPET demonstrates consistently faster time-to-accuracy (Coleman et al., 2019) compared to Mixed Precision across the benchmarks. Additionally, while Mixed Precision has already reached its limit by using FP16 on FP16-native GPUs, the 8-, 12-, 14- and 16-bit fixed-point computations enabled by MuPPET leave space for further potential speedup when targeting next- and current-generation (Fowers et al., 2018) precision-optimised fixed-point platforms. Similar to the analysis in Section 4.2, Micikevicius et al. (2018) and Wang et al. (2018) compare their schemes to baseline

FP32 training performed by them. The reported results demonstrate that their methods achieve similar accuracy results to our method by lying close to the respective FP32 training accuracy. As Wang et al. (2018) do not provide any results in terms of gains in wall-clock times and since they use custom FP8 hardware, their work could not be directly compared to our method.

|  | FP32 (Baseline) | Mixed Prec (Micikevicius et al., 2018) | MuPPET (Current Impl.) | MuPPET (Ideal) |
|---|---|---|---|---|
| **AlexNet** | 30:13 (1×) | 29.20 (1.03×) | 23:52 (1.27×) | 20:25 (1.48×) |
| **ResNet18** | 132:46 (1×) | 97:25 (1.36×) | 100:19 (1.32×) | 92:43 (1.43×) |
| **GoogLeNet** | 152:28 (1×) | 122:51 (1.24×) | 122:13 (1.25×) | 82:38 (1.84×) |

Table 2: Wall-clock time (GPU hours:mins) & relative acceleration for networks targeting ImageNet

### 4.4 PRECISION SWITCHING

To evaluate the ability of MuPPET to effectively choose an epoch to switch precision at, AlexNet and ResNet20 were first trained using MuPPET on the CIFAR-100 dataset. The hyper-parameters for MuPPET were kept the same across all runs. From the results it was noted that training at reduced precision and not switching at all causes a drop in validation accuracy of 1.4% and 1.3% for AlexNet and ResNet20 respectively, hence demonstrating the need to switch precisions when training at bit-widths as low as 8-bit fixed-point.

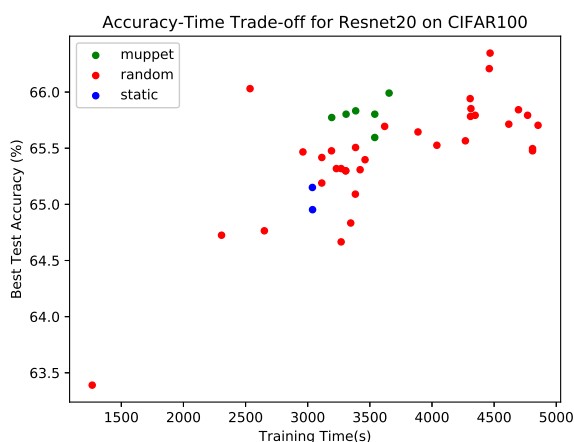

Figure 3: Accuracy vs time trade-off for ResNet20 MuPPET runs on CIFAR-100.

To demonstrate the benefits of a precision switching methodology, two further sets of experiments were conducted on ResNet20 using CIFAR100 as depicted in Fig. 3. First, 34 training runs were performed (34 red dots in Fig. 3), where for each training four epochs along the standard training duration were randomly selected and used as the switching points. Second, the switching strategy MuPPET generated for AlexNet and GoogLeNet was applied to ResNet20 (2 blue dots in Fig. 3). Fig. 3 shows the best test accuracy achieved by each of the runs and the training time as estimated by our performance model described in Sec. 4.3. It shows that for a given time-budget, MuPPET runs (6 green dots) outperform on average all other experiment sets, demonstrating the need for a precision switching policy that is real-time and agnostic to network and dataset in order to achieve a good accuracy-to-training-time trade-off.

## 5 CONCLUSION

This paper proposes MuPPET, a novel low-precision CNN training scheme that combines the use of fixed-point and floating-point representations to produce a network trained for FP32 inference. By introducing a precision-switching mechanism that decides at run time an appropriate transition point between different precision regimes, the proposed framework achieves Top-1 validation accuracies comparable to that achieved by state-of-the-art FP32 training regimes while delivering significant speedup in terms of training time. Quantitative evaluation demonstrates that MuPPET's training strategy generalises across CNN architectures and datasets by adapting the training process to the target CNN-dataset pair during run time. Overall, MuPPET enables the utilisation of the low-precision hardware units available on modern specialised processors, such as next-generation GPUs, FPGAs and TPUs, to yield improvements in training time and energy efficiency without impacting the resulting accuracy. Future work will focus on applying the proposed framework to the training of LSTMs, where the training process is more sensitive to gradient quantisation, as well as on the extension of MuPPET to include batch size and learning rate as part of its hyperparameters. Furthermore, we will explore improved quantisation techniques that could enable training convergence for bitwidths even lower than 8-bit fixed-point.

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

## A    APPENDIX A

The graphs in Fig. 4, 5 and 6 demonstrate both the training and validation loss of AlexNet, ResNet18 and GoogLeNet for MuPPET and FP32 runs on the ImageNet dataset. For each graph, the light gray lines indicate the point at which precision was switched in the MuPPET run. The green lines are used to show MuPPET behaviour and blue to show FP32 behaviour. Solid lines show validation loss and dashed lines show training loss.

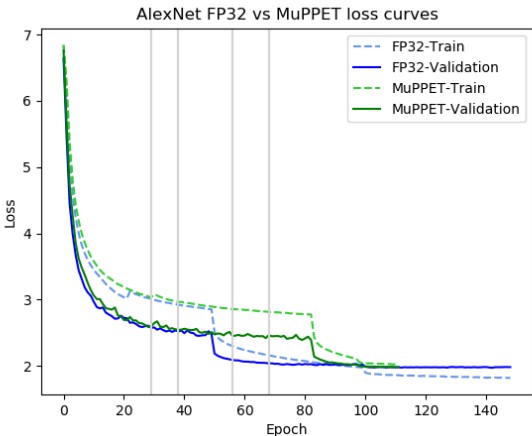

Figure 4: AlexNet training and validation loss values for FP32 and MuPPET on ImageNet.

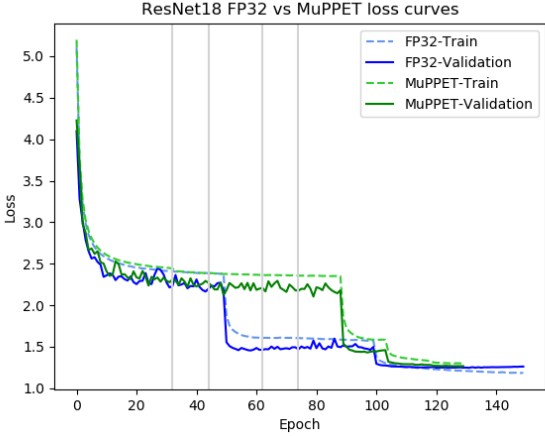

Figure 5: ResNet18 training and validation loss values for FP32 and MuPPET on ImageNet.

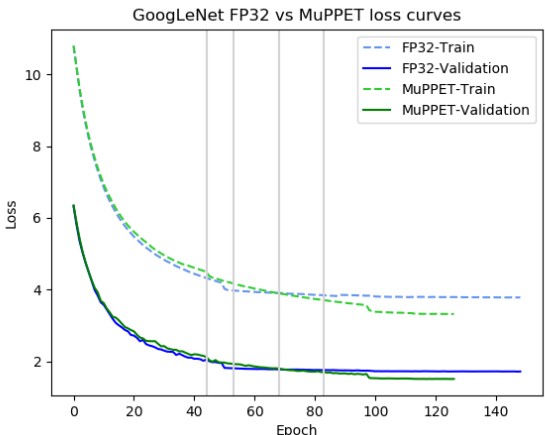

Figure 6: GoogLeNet training and validation loss values for FP32 and MuPPET on ImageNet.

## B  APPENDIX B

This section contains the larger versions of all the figures in the paper for enhanced clarity.

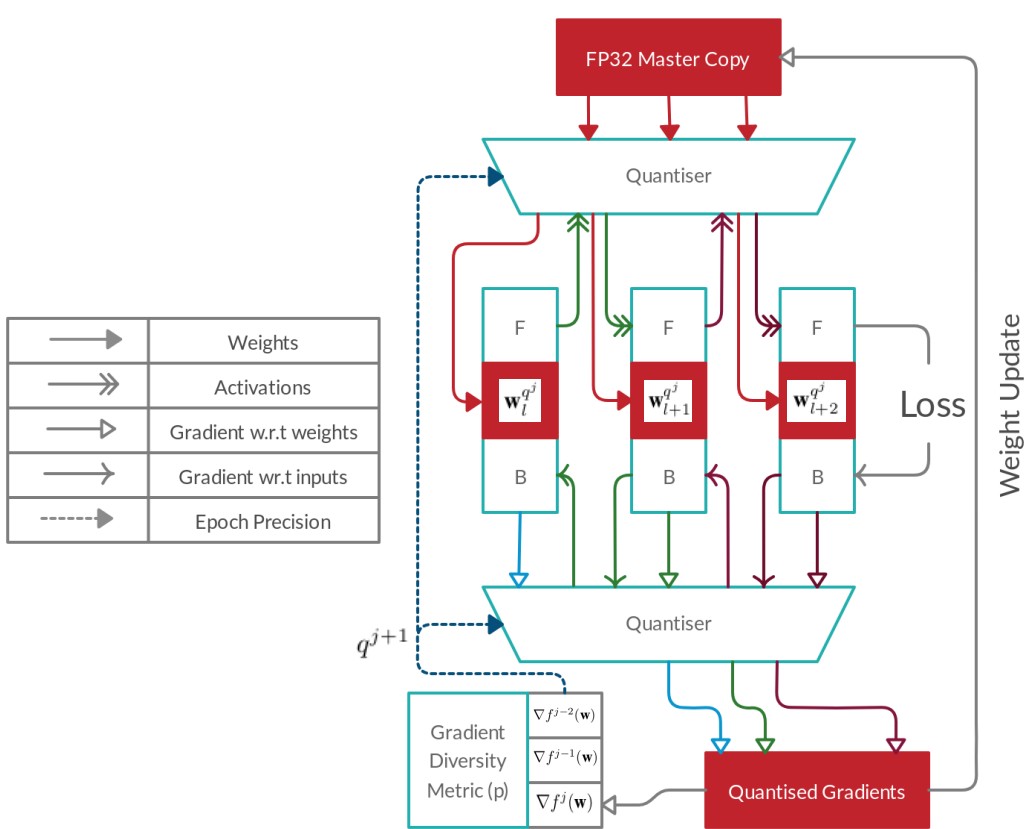

Figure 7: Precision-switching training scheme of MuPPET.

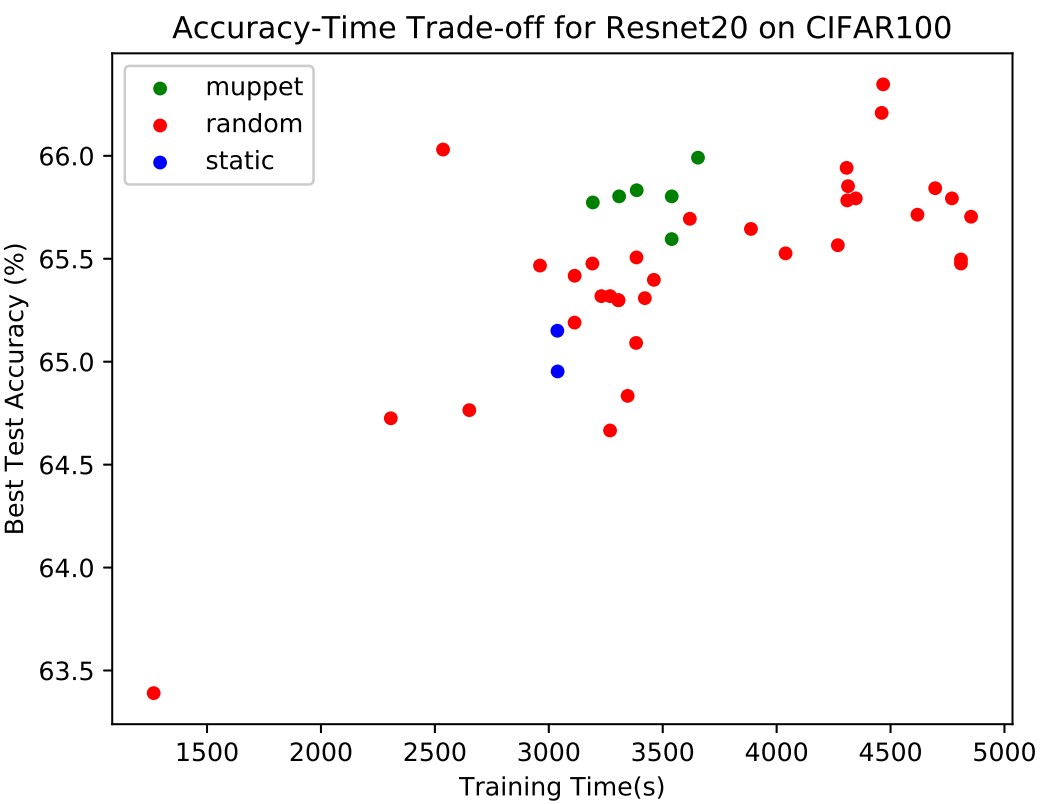

Figure 8: Accuracy vs time tradeoff for ResNet20 MuPPET runs on CIFAR100 - Final Test Accuracy

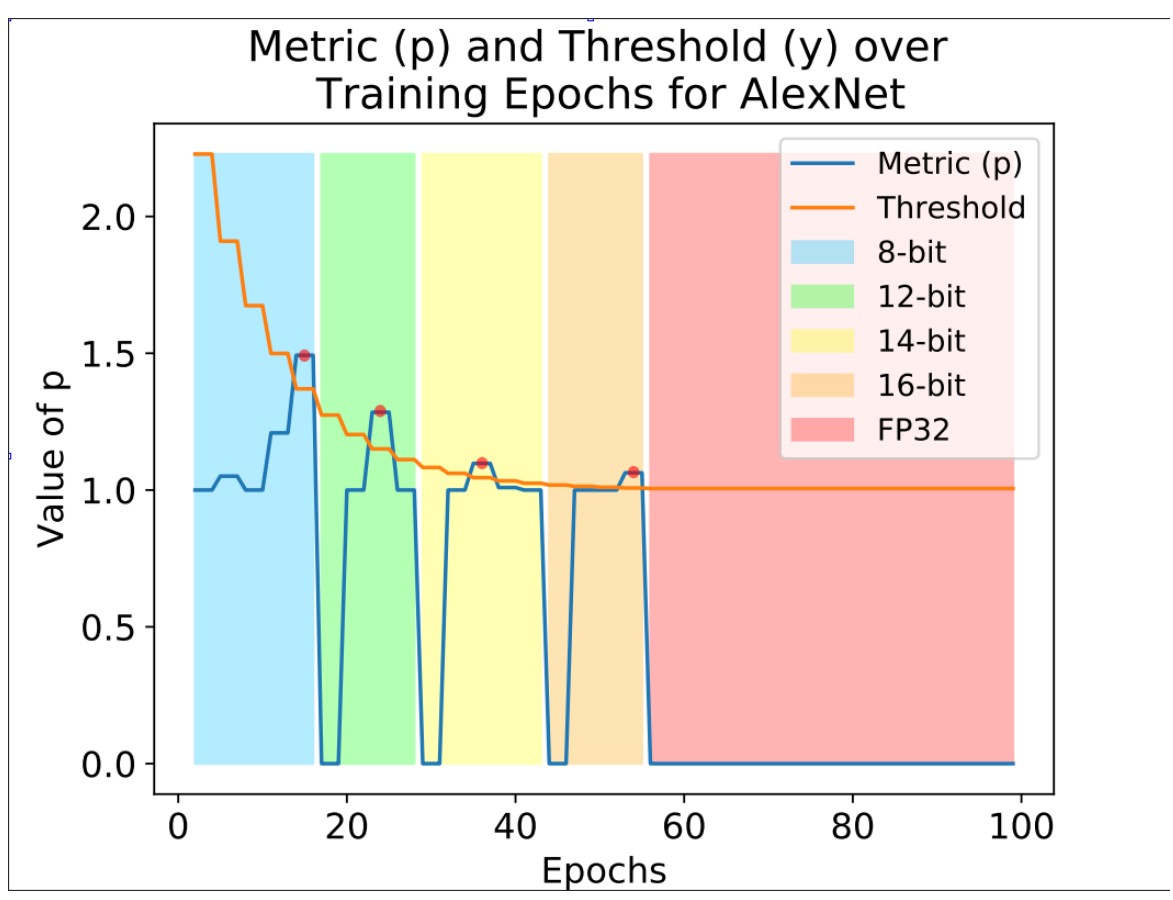

Figure 9: AlexNet CIFAR10 - Demonstration of the generalisability of $p$ over networks, datasets and epochs.

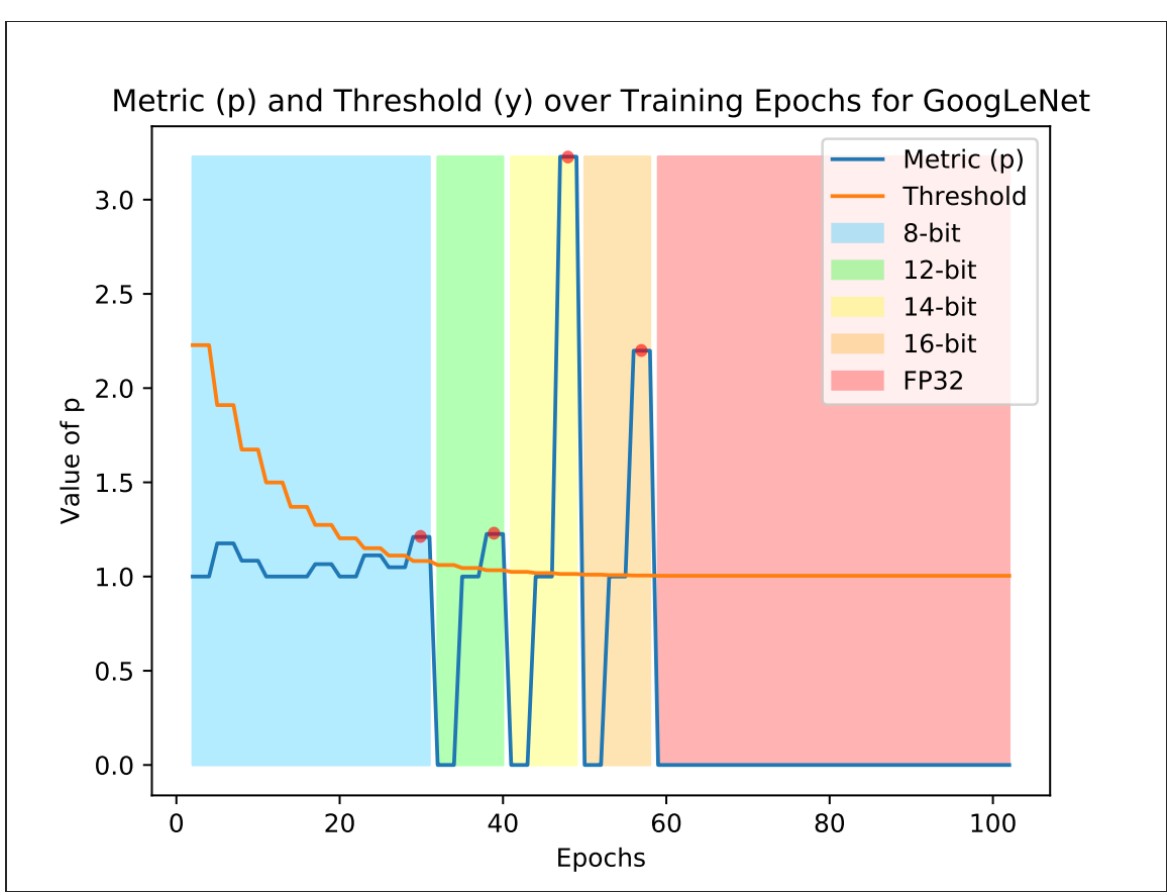

Figure 10: GoogLeNet CIFAR10 - Demonstration of the generalisability of $p$ over networks, datasets and epochs.

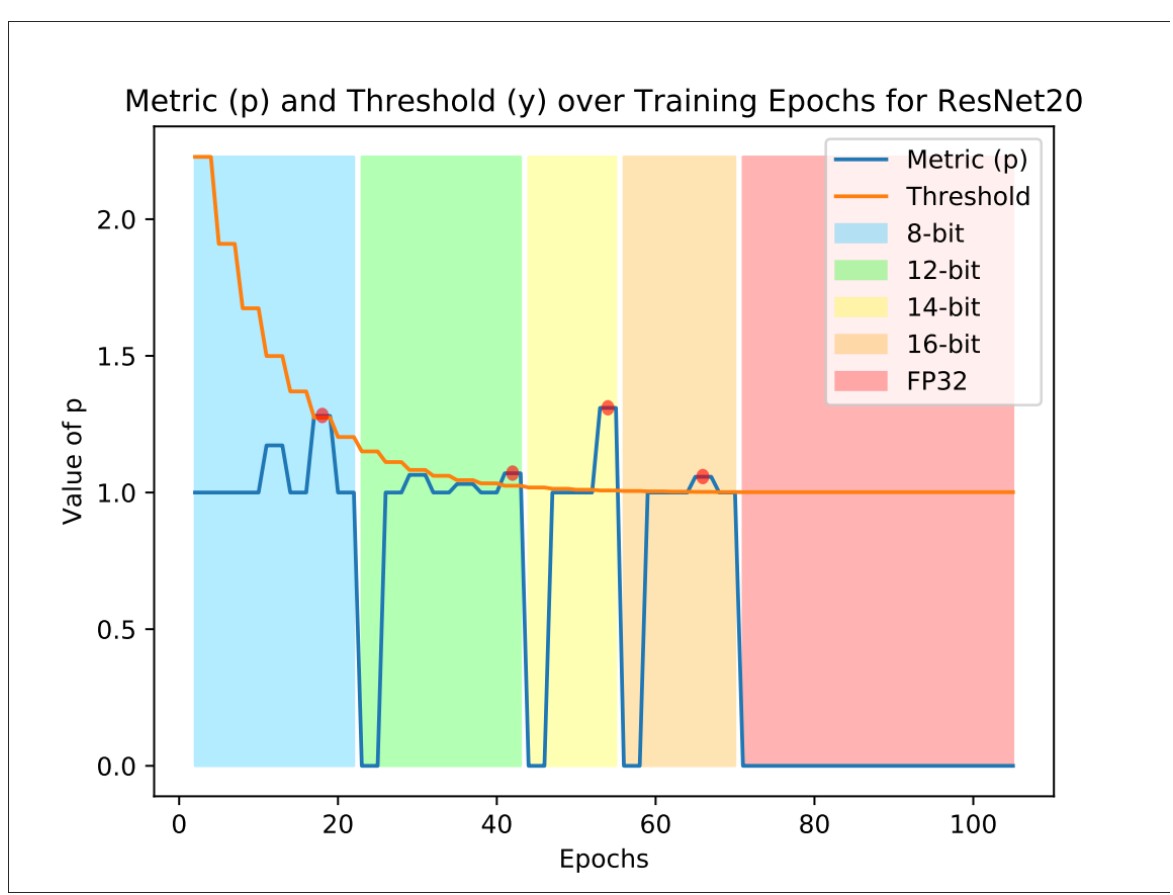

Figure 11: ResNet20 CIFAR10 - Demonstration of the generalisability of $p$ over networks, datasets and epochs.

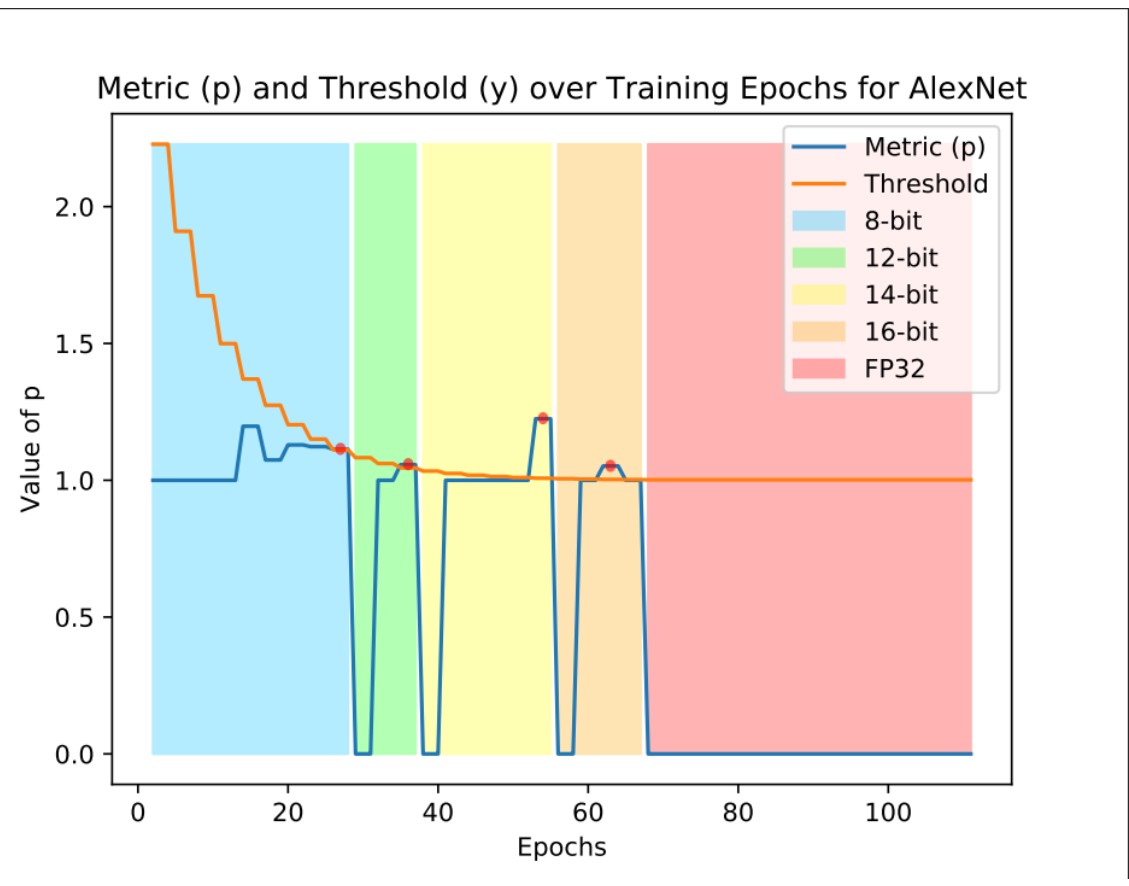

Figure 12: AlexNet ImageNet - Demonstration of the generalisability of $p$ over networks, datasets and epochs.

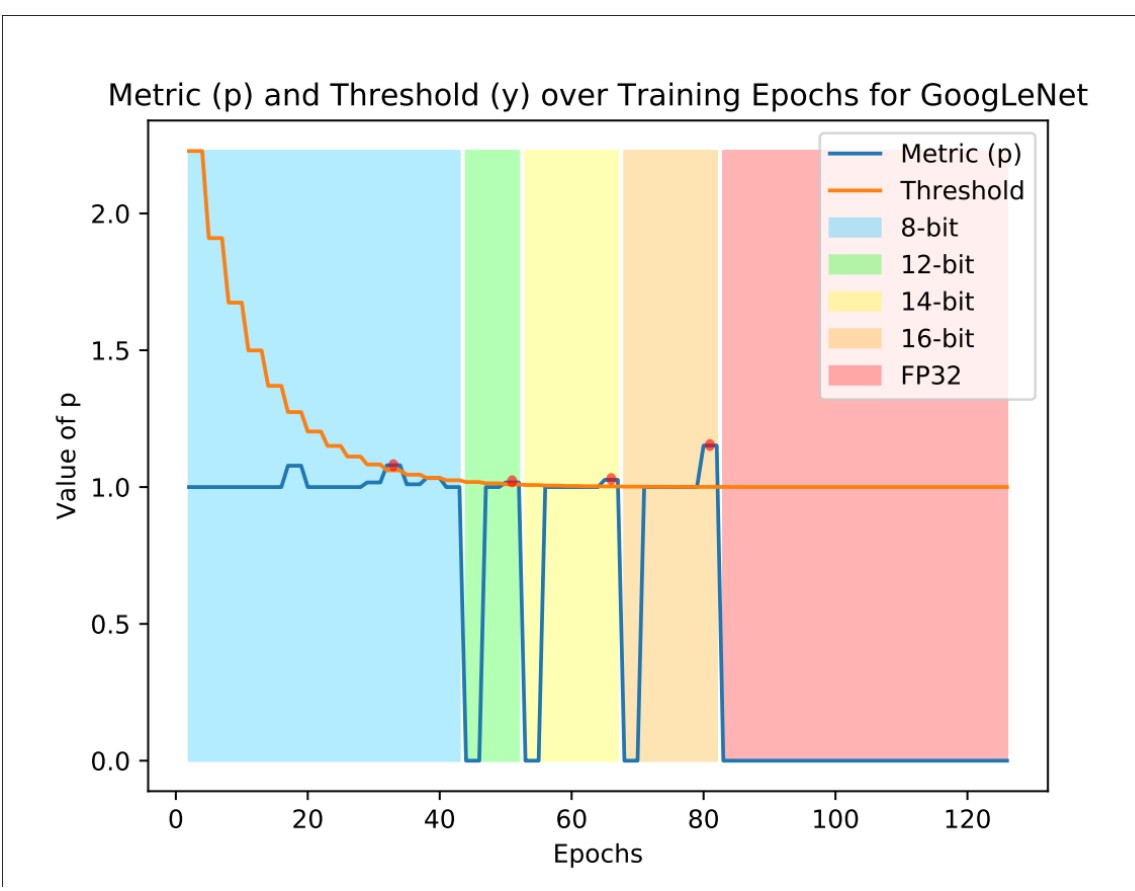

Figure 13: GoogLeNet ImageNet - Demonstration of the generalisability of $p$ over networks, datasets and epochs.

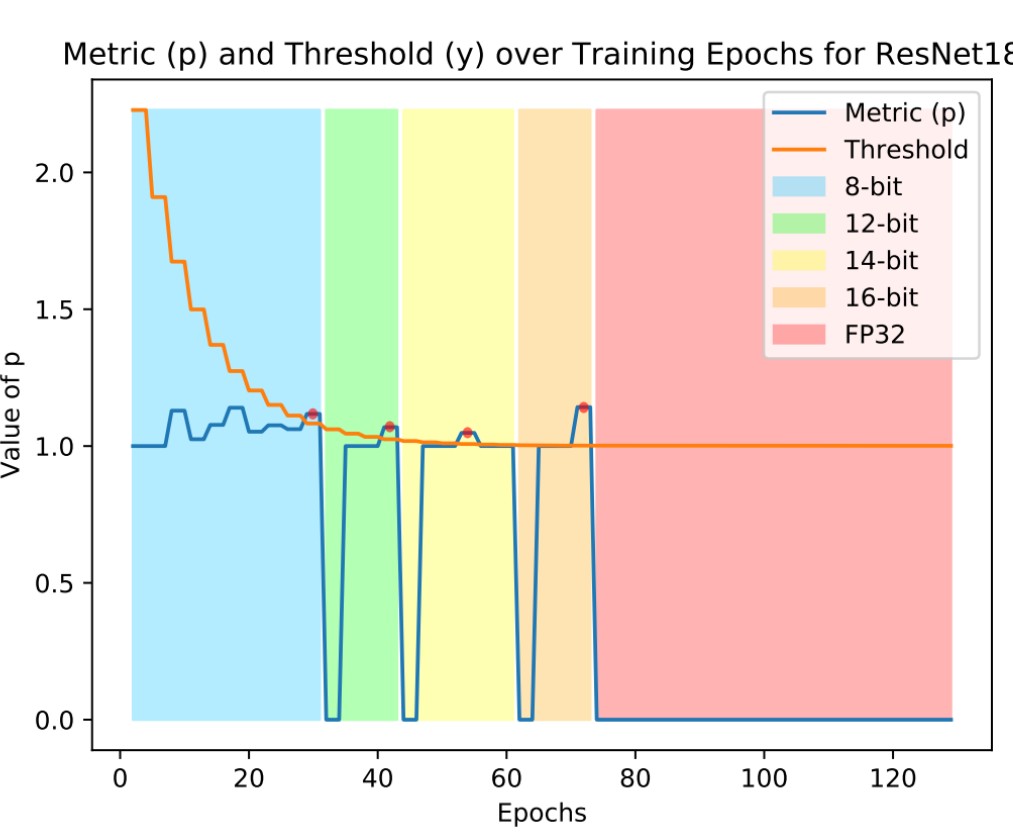

Figure 14: ResNet18 ImageNet - Demonstration of the generalisability of $p$ over networks, datasets and epochs.

