# OpenReview forum: "Multi-Precision Policy Enforced Training (MuPPET) : A precision-switching strategy for quantised fixed-point training of CNNs"
_ICLR.cc/2020/Conference — Reject_

### Official Review · AnonReviewer1 · 2019-10-23
**Official Blind Review #1**

**Rating:** 3

**Review:**

The article presents an approach to reduce the precision of weights, activations and gradients to speed up the training of deep neural networks. The precision of these values is increased according to a dynamic schedule such that the original classification accuracy is reached after training.

The manuscript is in most parts well written and the addressed topic is of general interest for the research community represented at ICRL. Still, I recommend a weak reject, since the core idea of the manuscript, i.e. the dynamic switching between precision levels, is not shown to be a necessary condition for good classification results.


Major points:
•	The introduction does not give a clear statement about the novel contribution of the paper. Only the very last paragraph is specific about the paper.
•	Your results support that step-wise increasing the resolution speeds up training without significant losses in accuracy. However, the impact of the gradient diversity, choice of p and threshold parameters on the performance of the trained networks are unclear. What is the isolated impact of every of these choices? According to Figure 2, pre-defined switching points between precision levels may also generalize between networks and datasets.
•	The description of the quantization scheme is not clear enough in order to reproduce the results:
o	Please give details about every step from FP32 to FPx values or cite appropriate literature.
o	Equation 4 and 5: How are the scaling factors SC determined?
o	Please clarify the difference/relation between n and WL.


Minor points:
•	Equation 3: What does “represent. range(q^i)” mean?
•	Text in Figure 1 and 2 is far too small and barely readable
•	Step 5 in Algorithm in Section 3.3: What does “p violates y more than gamma times” mean? What is y?
•	Please clarify “distribution approach”. Distribution of what?
•	Table 1: For the baseline experiments, the precision is switched from 8 to 32 bits, for MuPPET from 8 to 12 bits (see main text). What is the motivation behind these different choices?
•	Do you use any type of data augmentation?
•	Table 3: Please clarify “theoretical limit”. Does this limit include 12 and 14 bit quantisation. What do you mean by “optimized quantization implementation” in main text?

**Experience Assessment:**

I have published one or two papers in this area.

**Review Assessment: Checking Correctness Of Derivations And Theory:**

I assessed the sensibility of the derivations and theory.

**Review Assessment: Checking Correctness Of Experiments:**

I carefully checked the experiments.

**Review Assessment: Thoroughness In Paper Reading:**

I read the paper at least twice and used my best judgement in assessing the paper.

---

> ### Author Response · Authors · 2019-11-15
> **Response**
>
> Thank you for your comments.
>
> Major Points:
> We would like to clarify that the core idea of the manuscript is not to argue that dynamic switching between precision levels is a necessary condition for good classification results, but rather that knowing when to switch the precision of the computations can bring runtime benefits to the training process. Towards this, we demonstrate that MuPPET leads to identification of points for precision switching that can enable the inclusion of extreme precision regimes for training that were not considered before that  lead to its acceleration with no loss in the final accuracy. The manuscript has been revised in points to better reflect the above. Furthermore, we have demonstrated that the framework is agnostic to network and dataset which allows for a single generalisable approach.
>
> Section 4.4 has been updated with Fig. 3 which is an accuracy-time trade-off plot. Fig. 3 shows that, for a given time-budget, MuPPET outperforms runs that have 1) randomly chosen switching points or 2) switching points borrowed between networks and datasets, thus justifying a need for a framework that adapts at run-time to both network and dataset.
>
> Similar to the tuning of hyperparameters for training CNNs in general, the hyperparameters for MuPPET (threshold parameters) were explored in an empirical manner. In the revised manuscript, the choice of p is addressed in the paragraph beginning  “As long as the gradients …” in Section 3.3 which discusses how these choices make p and hence MuPPET agnostic to dataset and networks. Furthermore, the paragraph beginning “The likelihood of observing r gradients…” in Section 3.3 discusses the reasoning behind using gradient diversity as part of the metric. Nonetheless, the key point to take away here is that the further tuning of the hyperparameters for p is not crucial for the performance of MuPPET as it’s generalisability across datasets and networks has been demonstrated through our results.
>
> Section 3.2.1 has been updated with Eq. (4), (5) and (6) to make the quantisation strategy more explicitly defined. Furthermore, following your suggestion, the introduction has been made more concise and further emphasis has been put on describing the novel contributions of MuPPET.
>
> Minor Points:
> Equation 3 has been made more explicit in terms of what the representable range of q^i means.
> All figures have been additionally added to Appendix B at a larger scale to make them more readable.
> This was a typo and has now been fixed.
> “Distribution approach” referred to how the framework distributed the computations across GPUs, but has now been removed due to space considerations as this detail was not essential to the underlying principles of MuPPET.
> Table 1 has been replaced with the new discussion in Section 4.4. It was originally there to motivate the need for precision switching.
> We used the standard process described in each model's implementation for data augmentation and preprocessing, such as scaling and cropping the input image, horizontal flipping with a probability of 50% and normalisation by subtracting each channel mean and dividing by the standard deviation.
> Clarification of “theoretical limit” has been addressed in Section 4.3. All timings include computations at 8-, 12-, 14- and 16-bit fixed-point.

---

### Official Review · AnonReviewer3 · 2019-10-23
**Official Blind Review #3**

**Rating:** 3

**Review:**

Summary:
This paper proposes a training strategy called Multi-Precision Policy Enforced Training(MUPPET). This strategy aims to reduce training time by low-precision data representation and computations during the training stage. According to the gradient diversity, the authors introduce a precision-switching mechanism which chooses the best epoch to increase the precision. The validation accuracy and training time across several networks and datasets are shown in the experiments. However, the results are not superior enough compared with the state-of-the-art.

My detailed comments are as follows.


Positive points:

1. This paper proposes a new reduced-precision training scheme to speed up training by progressively increasing the precision of computations from 8-bit fixed-point to 32-bit floating-point. This scheme moves to reduced-precision fixed-point computations while updating an FP32 model in order to push the boundaries of reduced-precision training.

2. The authors propose a metric to decide when to switch the precision inspired by gradient diversity introduced by [1]. In this paper, the gradient diversity is enhanced by considering gradients across epochs instead of mini-batches. The proposed metric can be seen as a proxy for the amount of new information gained in each training step. Therefore, the metric can decide the most appropriate epoch at run time to increase the precision.

3. The proposed low-precision CNN training scheme is orthogonal and complementary to existing low-precision training techniques.




Negative points:

1. The proposed approach does not match the description in this paper. The authors describe “This approach enables the design of a policy that can decide at run time the most appropriate quantization level for the training process”. In fact, this approach just decides which epoch to increase the quantization level while the levels of quantized precisions are fixed, rather than deciding the most appropriate quantization level.

2. The setting of quantized precision levels (8-, 12-, 14- and 16-bit precisions) is confusing. Please illustrate how to choose the number of quantized bit and the number of quantized precision levels.

3. The presentation of the precision switching policy is confusing and the notations are unclear. For example, in section 3.3, the ratio “p” needs more description because it is a key value in the policy, but lacks an explanation in this section. So please explain more about the motivation of ratio “p” in this section. 	In section 3.3, in step 5 of the proposed precision switching policy, the authors do not explain the meaning of “y”.

4. In figure 2, the precision switch is not triggered even though the value of p violates the threshold more than 2 times, which mismatches the description in section 3.3.

5. The proposed strategy has no obvious advantages. There are some scenes that the proposed strategy does not perform well. For example, the Top-1 validation accuracy on ImageNet of AlexNet and ResNet with MuPPET strategy is much lower than FP32 baseline. Compared with [2], the proposed method is more complex but not superior enough.

6. The authors do not show the training and validation curves. However, the training and validation curves are common used to show more details of the training process, such as in [2] and [3]. Please show and analyze the training and validation curves of the proposed scheme and the baseline.


Minor issues:
Some spelling and grammar mistakes.


Reference：
[1] Dong Yin, Ashwin Pananjady, Max Lam, Dimitris Papailiopoulos, Kannan Ramchandran, and Peter Bartlett. Gradient Diversity: a Key Ingredient for Scalable Distributed Learning. In 21st International Conference on Artificial Intelligence and StatiZZstics (AISTATS), pp. 1998–2007, 2018.
[2] Paulius Micikevicius, Sharan Narang, Jonah Alben, Gregory Diamos, Erich Elsen, David Garcia, Boris Ginsburg, Michael Houston, Oleksii Kuchaiev, Ganesh Venkatesh, and Hao Wu. Mixed Precision Training. In International Conference on Learning Representations (ICLR), 2018.
[3]  Suyog Gupta, Ankur Agrawal, Kailash Gopalakrishnan, and Pritish Narayanan. Deep Learning with Limited Numerical Precision. In 32nd International Conference on Machine Learning (ICML), pp. 1737–1746, 2015.


**Experience Assessment:**

I have read many papers in this area.

**Review Assessment: Checking Correctness Of Derivations And Theory:**

I carefully checked the derivations and theory.

**Review Assessment: Checking Correctness Of Experiments:**

I carefully checked the experiments.

**Review Assessment: Thoroughness In Paper Reading:**

I read the paper thoroughly.

---

> ### Author Response · Authors · 2019-11-15
> **Response**
>
> Thank you for your review.
> Regarding point 1), we have edited the sentence towards the end of the introduction to better phrase the impact and purpose of this work.
>
> Regarding point 2), it has been added to Section 3.3.1 that these quantisation levels were empirically chosen. The reasoning behind this is that we want to increase the utilised word length as little as possible in order to gain the most performance (runtime) from the computation platform, but moving too little will not result in “enough” information gain and will force the system to switch regimes too often leading to the waste of computational resources.
>
> Regarding point 3), Section 3.3 has been updated to address all the mentioned points.
>
> Regarding point 4), this never occurs, however, Fig. 2 will be updated to highlight the exact points at which the threshold is violated. A short discussion has also been added in Sec.4.1 to clearly indicate switching points.
>
> Regarding point 5), we feel that a difference of < 1% in Top-1 Validation Accuracy on ImageNet is not considered “much lower” and fluctuations at these levels can be seen between identical training runs. Furthermore, with respect to GoogLeNet, for the exact same hyperparameters, we achieve a +4.55% improvement in validation accuracy which is significant. Compared to [2], as has been added to the discussion in Section 4.3, this work pushes this boundary even further and opens the possibility for performing computations at wordlengths much lower than 16-bit, and at fixed-point instead of floating-point. With the availability of native hardware (e.g. 8-bit fixed-point computations in NVIDIA’s Turing GPUs), being able to perform training at these precisions without compromising accuracy (as shown in this paper) carries significant advantages.
>
> Regarding point 6), we have added these graphs to Appendix A with a short description of what they show.

---

### Official Review · AnonReviewer2 · 2019-10-24
**Official Blind Review #2**

**Rating:** 3

**Review:**

Overall an interesting paper, though I wished a more detailed presentation of the reasoning behind the algorithm would have been provided. As it stands it feels a bit heuristic.

In particular I don't understand the motivation between the switching mechanism. Basically it says if the gradients are co-aligned between epochs it means there is not much to learn anymore!? Why? Intuitively if the gradients would go to 0 or become very small maybe you would want to increase precision. Or if you have high variance you could argue that the expected gradient would be 0 and hence you are not really making progress, i.e. you are just moving left-right. But if all gradients agree on a moving direction, why is that a bad thing? I know the heuristic is borrowed from a different work, but since it feels as such an integral part of MuPPET I think you should explain it better.

I guess a few details about the algorithm as well. When you say you look at the diversity of the gradients over the epochs, is this the batch gradient !?

There are some small typos (e.g. FP23 instead FP32).

I find the justification for AlexNet to be adhoc (it switched at the wrong time, but that allowed to take more advantage of computation in the low precision hence it was faster). The switching mechanism should only care of when the gradients are not informative anymore, not how much compute you are wasting .

**Experience Assessment:**

I have published one or two papers in this area.

**Review Assessment: Checking Correctness Of Derivations And Theory:**

I did not assess the derivations or theory.

**Review Assessment: Checking Correctness Of Experiments:**

I assessed the sensibility of the experiments.

**Review Assessment: Thoroughness In Paper Reading:**

I made a quick assessment of this paper.

---

> ### Author Response · Authors · 2019-11-15
> **Response**
>
> Thank you for your review. Further details on the motivation behind the switching mechanism has been added to Section 3.3 in the revised version of the paper, particularly at the paragraph beginning “The likelihood of observing r gradients…”.
> Additionally we would like to state that when the observed p-value is high, this could be due to either true co-alignment of the gradients, or due to information loss from quantisation the gradients appear to be co-aligned. We find that it is unlikely to observe multiple minibatches across 3 epochs to have similar gradients. As a result, seeing this behavior would indicate that information is being lost through quantisation producing the observed low gradient diversity.
>
> The gradients being used are those obtained from the last minibatch of each epoch. This has been updated in point 1) of Section 3.3.
>
> It was not our intention in the original version for it to sound as an adhoc justification of MuPPET for AlexNet, more so just an observation of the experiment that we ran. However, Section 4.4 has been revised with more thorough and appropriate experiments and analysis in the current version of the paper.

---

### Decision · Program_Chairs · 2019-12-19

**Decision:**

Reject

**Comment:**

The submission presents an approach to speed up network training time by using lower precision representations and computation to begin with and then dynamically increasing the precision from 8 to 32 bits over the course of training. The results show that the same accuracy can be obtained while achieving a moderate speed up.

The reviewers were agreed that the paper did not offer a signficant advantage or novelty, and that the method was somewhat ad hoc and unclear. Unfortunately, the authors' rebuttal did not clarify all of these points, and the recommendation after discussion is for rejection.